# Receptor for Advanced Glycation End Products (RAGE): A Pivotal Hub in Immune Diseases

**DOI:** 10.3390/molecules27154922

**Published:** 2022-08-02

**Authors:** Qing Yue, Yu Song, Zi Liu, Lin Zhang, Ling Yang, Jinlong Li

**Affiliations:** 1Hebei Key Laboratory for Organ Fibrosis Research, School of Public Health, North China University of Science and Technology, Tangshan 063210, China; yueqing0705@163.com (Q.Y.); songyujiayou@163.com (Y.S.); lz13184993585@163.com (Z.L.); 15553806071@163.com (L.Y.); 2Department of Internal Medicine Nursing, School of Nursing, Wannan Medical College, 22 Wenchang West Road, Higher Education Park, Wuhu 241002, China; yaoran2008@163.com

**Keywords:** advanced glycation end-product receptor, immune, high-mobility group protein 1, nuclear factor kappa-B

## Abstract

As a critical molecule in the onset and sustainment of inflammatory response, the receptor for advanced glycation end products (RAGE) has a variety of ligands, such as advanced glycation end products (AGEs), S100/calcium granule protein, and high-mobility group protein 1 (HMGB1). Recently, an increasing number studies have shown that RAGE ligand binding can initiate the intracellular signal cascade, affect intracellular signal transduction, stimulate the release of cytokines, and play a vital role in the occurrence and development of immune-related diseases, such as systemic lupus erythematosus, rheumatoid arthritis, and Alzheimer’s disease. In addition, other RAGE signaling pathways can play crucial roles in life activities, such as inflammation, apoptosis, autophagy, and endoplasmic reticulum stress. Therefore, the strategy of targeted intervention in the RAGE signaling pathway may have significant therapeutic potential, attracting increasing attention. In this paper, through the systematic induction and analysis of RAGE-related signaling pathways and their regulatory mechanisms in immune-related diseases, we provide theoretical clues for the follow-up targeted intervention of RAGE-mediated diseases.

## 1. Introduction

Receptor for advanced glycation end products (RAGE) was initially considered the only protein that can bind to advanced glycation end products (AGEs) and regulate some signal pathways. As a transmembrane receptor, the structure of RAGE is divided into extracellular, transmembrane, and intracellular segments and exists in vivo as transmembrane molecules and soluble molecules [1]. Because of its unique structure and existing form, RAGE can bind to various ligands, such as S100, calcium/grain, high-mobility group protein 1 (HMGB1), and amyloid β-protein (Aβ) [2]. Soluble receptor for advanced glycation end products (sRAGE) does not have a signal transmission function due to its lack of transmembrane and intracellular segments [3]. sRAGE can competitively bind to RAGE ligands, thus antagonizing the pathological effects mediated by RAGE [4]. 

Studies have revealed that high-sugar diets, amyloidosis, oxidative stress, and other unique environments can significantly induce RAGE expression on the surface of smooth muscle cells, neurons, and other cells. RAGE participates in critical physiological processes, such as regression of inflammation, maintenance of cell homeostasis, and postinjury repair and regeneration [5]. For example, a low concentration of S100B regulates cell proliferation and differentiation through RAGE under physiological conditions. Under pathological conditions, the combination of S100B and RAGE stimulates the release of proinflammatory cytokines, such as TNF-α, and triggers MAPK and NF-κB signals [6]. RAGE has become a vital regulator of the innate immune response. In the context of chronic inflammation, the upregulation of RAGE signals causes pathological changes. Advanced oxidation protein products (AOPPs) are critical in the development of various skin diseases. A series of studies has shown that RAGE is the receiver of AOPPs, which also play an indispensable role in the development of immune diseases, such as psoriasis and systemic lupus erythematosus [7]. Current research has increased attention on the potential of RAGE and its ligands to target human-related diseases [8,9,10]. Many studies have expounded on the disease diagnostic potential of RAGE and its ligands as biomarkers and used them to study the pathological progress of diseases and evaluate the severity of conditions [11,12,13]. However, the regulatory mechanism of RAGE signal transduction and its role in immune-associated diseases have not been fully elucidated. Therefore, research on the RAGE-related signaling pathway and its role in immune disorders must be further summarized and analyzed.

## 2. RAGE Structure

RAGE consists of extracellular, hydrophobic, transmembrane, and intracellular segments. The extracellular V-shaped region provides RAGE–ligand binding sites. Intracellular fragments can bind to various intracellular signal molecules and mediate signal transduction to cause cascade reactions, including full-length type, truncated C-terminal type, and truncated N-terminal type. The truncated C-terminal type is endogenous secretory soluble RAGE (esRAGE), which can be secreted by cells and contains only the extracellular segment. In contrast, the truncated N-terminal type consists of the transmembrane region and an intracellular component. Membrane-associated proteases can remove the transmembrane component of RAGE by hydrolysis, and the released extracellular segment can form soluble sRAGE with esRAGE. sRAGE can competitively bind to RAGE ligands, but binding to ligands terminates intracellular signal transduction due to the loss of transmembrane and intracellular fragments [14] (Figure 1).

### 2.1. RAGE Ligand

RAGE is composed of polymorphic domains (such as V, C1, and C2), gene polymorphisms (such as rs1800624 and rs2071288), and isomers (courage and esRAGE) and can bind to various endogenous and exogenous ligands [15,16]. Its ligands include AGEs, high-mobility group protein 1, S100 protein family members, amyloid β peptide, type I collagen, and type IV collagen. It also provides surface molecules for prions, bacteria, and lymphocytes [17,18,19]. After binding the RAGE ligand to the membrane receptor RAGE, it can activate RAGE and mediate various signal transduction processes, making RAGE–ligand binding a complex process.

### 2.2. AGEs

AGEs can produce brown and specific fluorescent molecules, are not easily degraded because of their stability to enzymes, and can act as ligands to interact with various cell-membrane-specific receptors to exert biological effects. Diets high in sugar, aging, and oxidative stress all increase the AGE levels in the body [20]. Under pathological conditions, AGEs can cause abnormal tissue structure and function, resulting in pathological cascade changes, which play an essential role in age-related degenerative diseases, such as Alzheimer’s disease, Parkinson’s disease, and atherosclerotic disease [21]. The combination of AGEs and RAGE can activate the downstream nuclear factor kappa-B (NF-κB) signaling pathway and promote the secretion of TNF-α, IL-1β, IL-6, and other cytokines, contributing to inflammation [22].

### 2.3. HMGB1

HMGB1, also known as nerve axon growth factor, is a rich non-histone chromosome protein that can be used as a transcription factor to regulate the expression of some genes and can also be used as a proinflammatory factor to cause inflammatory diseases [23]. It has been reported that in the early stage, the combination of HMGB1 and RAGE can cause the outward growth of developing embryonic nerve processes, with a binding ability seven times higher than that of AGEs [24]. Follow-up studies revealed that it can also be secreted from cells as an intercellular messenger factor. When stimulated, HMGB1 is released from the cell through the acetylation of lysine residues. Dendritic cells, macrophages, and natural killer cells can actively secrete HMGB1 [25]. When it is released actively or passively, it can accumulate white blood cells, trigger the release of inflammatory factors, and stimulate the inflammatory response [26].

### 2.4. Members of the S100 Protein Family

The calpain family is a group of calcium-binding functional proteins. It was initially extracted from the brain of a cow. The S100 protein family is small in relative molecular weight and composed of two distinct EF hands flanked by hydrophobic regions at either terminus and separated by a central hinge region. The carboxy-terminal EF hand is usually referred to as the canonical Ca^2+^-binding loop and encompasses 12 amino acids. In contrast, the amino-terminal loop comprises 14 amino acids and has a lower affinity for Ca^2+^. It exhibits intracellular and extracellular regulatory activity and participates in various cellular activities. S100 protein contains 25 members. Theoretically, the interaction between S100 protein and RAGE is a common feature of the S100 family, but it has been reported that only some of the members are ligands of RAGE [27]. The identical ligands that interact with RAGE are S100A4, A6, A7, A11, A12, A13, A15, B, P, and two possible ligands: S100A1 and S100A8/9 [28]. S100B is a small soluble protein that is mainly secreted by astrocytes in the central nervous system and plays a vital role in transmitter secretion, structure maintenance, information transmission, energy metabolism, and information transmission.

### 2.5. Other RAGE Ligands

Other ligands that bind to and interact with RAGE include β-amyloid, collagen I, and collagen IV [29]. After attaching to extracellular matrix components, such as collagen, RAGE can regulate the dispersion of lung angiotensin type 1 (AT1) stem cells. In addition, RAGE binds to soluble Aβ, which promotes oxidative damage, the release of inflammatory cytokines, and the formation of central plaques, which aggravates the progression of Alzheimer’s disease [30].

Owing to its ability to interact with different ligands, RAGE activates a variety of intracellular signaling pathways, such as p38MAPK, protein kinase B (Akt), extracellular regulated protein kinases (ERKs), mammalian transparent 1 (mDia1), and Rho GTPase (Rac1, Cdc42), which activate cascade transcription factors, such as NF-κB, SP-1, signal transducer and activator of transcription 3, (STAT3), and early growth response protein 1 (EGR-1) [1,31,32]. Due to its participation in many complex pathways, RAGE is a pivotal hub in immune diseases such as systemic lupus (SLE), rheumatoid arthritis (RA), Alzheimer’s disease (AD), and cancer, as well as in physiological processes, such as cellular aging and autophagy.

## 3. RAGE Regulates MAPK/NF-κB Signaling Pathway and Its Role in Immune-Associated Disease

As an essential protein complex, NF-κB can be found in almost all animal cells. It can affect cell growth, differentiation, and apoptosis by controlling critical physiological processes, such as DNA transcription and cytokine production. In resting cells, NF-κB usually binds to its protein inhibitor of NF-κB (IκB). Therefore, it is in a state of inhibition and does not exert the function of transcriptional activity. After activation, the MAPK pathway can hydrolyze IκB and dissociate it from NF-κB. The free NF-κB enters the nucleus to regulate the transcriptional expression of inflammatory factors [33]. The face of RAGE on the cell membrane is at a low level under normal physiological conditions. When the body’s inflammatory factors increase, the body is traumatized or suffers from abnormal conditions, such as diabetes. The content of ligands, such as AGEs and HMGB1, increase, and the expression of RAGE is upregulated [1]. When RAGE is combined with ligands, it promotes the downstream phosphorylation of the p38 MAPK protein, thus increasing the expression of the NF-κB signaling path [34]. NF-κB encourages the expression of inflammatory factors, such as TNF-α and IL-1β, by regulating the target genes and triggering related inflammatory and autoimmune responses, resulting in persistent tissue damage [35]. When NF-κB is activated, it can promote the binding of RAGE to its ligand and further encourage the continuous expression of cytokines and tissue factors [36] (Figure 2).

Table 1 summarizes the related ligands and mechanisms of the RAGE-MAPK/NF-κB pathway in SLE, RA, pulmonary fibrosis, and AD. 

### 3.1. RAGE Regulates MAPK/NF-kB Signaling Pathway and Its Role in Mediating Systemic Lupus Erythematosus

SLE is an autoimmune disease involving various systems and organs, with diverse and variable clinical manifestations. The pathogenesis of SLE is complicated. The interaction between genetic and environmental factors, the production of pathogenic antibodies, and the deposition of immune complexes (ICs) formed by combining autoantibodies and antigens are all causes of SLE [50]. HMGB1 binds to Toll-like receptors, such as TRL-2, TRL4, and RAGE, to regulate the immunoinflammatory response by mediating the MAPK/NF-κB signal pathway. In addition, it has been found that the immune complex formed by the combination of HMGB1 and DNA plays a vital role in the pathogenesis of SLE [51,52]. These findings suggest that HMGB1 plays a crucial role in the occurrence and development of SLE.

To confirm the above conjecture, some studies have further examined the level of HMGB1 in SLE patients and animal models. The results showed that the expression level of HMGB1 in patients and animals increased, and the inflammatory response of related immune cells increased significantly, which led to the occurrence of SLE [37]. Immune complexes can activate the expression of RAGE in human endothelial cells, participate in the response of the HMGB1-RAGE axis in promoting SLE vasculitis, and induce the production of TNF-α and B-cell-activating factors. Some studies have revealed that plasma HMGB1 levels in patients with SLE are significantly higher than those in healthy subjects and positively correlated with the concentration of plasma antinuclear antibody [53]. In addition, the concentration of AGEs in skin cells and serum, as well as the expression of RAGE mRNA in peripheral blood monocytes, are significantly increased in patients with SLE. Immunohistochemical detection revealed almost no expression of RAGE in normal glomerular cells of normal subjects. However, prominent expression still occurs in glomeruli patients with SLE [38]. In summary, we can speculate that the possible mechanism of RAGE involves binding to ligands and activation of transcription factor NF-κB through the MAPK pathway to enhance inflammatory response and promote SLE. At present, direct evidence about the involvement of RAGE in SLE is rare, whereas the role of its ligands in SLE is more prominent; therefore, the mechanism of RAGE in SLE requires further investigation.

### 3.2. The Role of the MAPK/NF-kB Signaling Pathway Regulated by RAGE in Mediating Rheumatoid Arthritis

RA is characterized by inflammatory synovitis, leading to joint injury and deformity. Although the formation of RA is not precise, the overexpression of many cytokines is a significant cause of bone destruction. The cytokines that play a critical role in this process are interleukin-1β (IL-1β) and TNF-α [54]. Previous studies have confirmed that HMGBl is highly expressed in the synovium of patients with RA, and anti-HMGB1 antibody can inhibit the inflammatory response caused by HMGB1, reduce the damage to joint tissue, and effectively alleviate arthritis caused by HMGB1 [55]. As one of the ligands with the highest affinity for RAGE, HMGB1 activates NF-κB mainly through an RAS/MAPK kinase cascade reaction after binding to each other and induces the production of a large number of cytokines and inflammatory factors [56].

Comparison of the expression of RAGE mRNA, TNF-α, and IL-1β in synovial tissue of patients with rheumatoid arthritis and osteoarthritis revealed that the concentration of TNF-α and IL-1β in synovial tissue of the former was higher than that of the latter. There was a significant positive correlation between the attention of two inflammatory factors and the expression of RAGE. These results suggest that HMGBl-RAGE can promote the expression of inflammatory cytokines by activating NF-κB [39]. It can be inferred that HMGBl-RAGE upregulates the expression of inflammatory factors in RA by activating NF-κB. In addition, the test revealed that patients positive for serum rheumatoid factor had lower serum sRAGE levels than patients with RA disease without a rheumatoid factor [57]; a number of studies have corroborated these results [58,59,60]. Recent studies have revealed that compared with healthy controls, RA patients have higher serum sRAGE levels, which is positively correlated with disease activity [61]. This is phenomenon is contrary to the results reported in previous studies. We speculate that the increase in serum sRAGE levels is most likely due to the decrease in proinflammatory RAGE–ligand binding when disease activity is controlled. High levels of sRAGE production and secretion may indicate the response of compensatory anti-inflammatory mechanisms to tissue damage by acting as bait receptors and proinflammatory ligands and preventing them from transmitting harmful signals. This theory further supports the hypothesis that decreased sRAGE levels in RA patients may increase inflammatory tendencies. In summary, HMGB1/RAGE can damage joints by regulating the MAPK/NF-κB signal pathway, resulting in RA-related symptoms. Furthermore, by competitively binding RAGE ligands, sRAGE inhibits the entry of inflammatory cells into the joint cavity (Figure 3).

### 3.3. The Role of the MAPK/NF-kB Signaling Pathway Regulated by RAGE in Pulmonary Fibrosis

When the lungs are injured, immune cells (such as monocytes, macrophages, T cells, B cells, and NK cells) and some non-immune cells (such as endothelial cells, epidermal cells, and fibroblasts) secrete a variety of inflammatory, profibrotic cytokines and chemokines, leading to persistent inflammation in the lungs and accumulation of extracellular matrix, ultimately leading to pulmonary fibrosis disease. In the pathogenesis of idiopathic pulmonary fibrosis, it is indispensable for epithelial cells to form myofibroblasts through epithelial–mesenchymal transition (EMT) [62]. RAGE was found to be significantly expressed on the membrane of alveolar macrophages, vascular smooth muscle cells, pulmonary endothelia, and epithelial cells [63,64]. Considerable evidence has shown that the interaction between RAGE and ligands is essential in pathological processes, such as inflammation and fibrosis [65,66]. S100A12 is reported to be highly expressed in the lungs of patients with acute respiratory distress syndrome, which activates pulmonary inflammation and endothelial cells through the binding of RAGE, leading to an increase in fibrogenic growth factor and the induction of epithelial–mesenchymal transformation [40,67]. A large amount of HMGB1 can induce EMT in normal alveolar type II epithelial cells, although it is difficult for alveolar type II epithelial cells without the RAGE genotype to produce EMT, suggesting that HMGB1–RAGE–ligand interaction can induce the EMT process of alveolar type II epithelial cells [68]. Through the stimulation of pulmonary interstitial cells in vitro, it was found that S100A9 stimulation of fibroblasts can significantly increase the expression of inflammatory factors in a dose-dependent manner. Treatment with ERK1/2MAPK inhibitors, anti-RAGE antibodies, and NF-κB inhibitors significantly downregulates S100A9-induced fibroblast proliferation and inflammatory cytokine secretion. It was confirmed that S100A9 and RAGE can activate fibroblasts through ERK1/2, MAPK, and NF-κB signaling pathways [41]. In summary, S100 protein RAGE can mediate the secretion of inflammatory factors and promote fibrosis through the MAPK pathway.

MAPK/NF-KB, as the first confirmed RAGE regulatory pathway, is also a classic inflammatory pathway. Whether AGEs, HMGB1, or S100 protein family members, RAGE classic ligands can activate the MAPK/NF-KB pathway and promote the occurrence and development of SLE, RA, pulmonary fibrosis, and other immune diseases.

## 4. The Role of RAGE-Related Signaling Pathways in Alzheimer’s Disease

### 4.1. β-Amyloid Protein Regulates the RAGE-Related Signaling Pathway and Its Role in Alzheimer’s Disease

AD is a neurodegenerative disease characterized by memory impairment, executive dysfunction, and personality and behavioral changes. In recent years, an increasing number of studies have found that AD is also an immune-related disease, and the inflammation caused by the brain immune system can drive the deterioration of AD. However, its pathogenesis has not been studied. The mainstream theory is that β-secretase and γ-secretase on the mass membrane are cut sequentially at the N and C ends of the amyloid prebiotic protein (APP) to form a β-amyloid peptide. The primary forms are Aβ40 and Aβ42 [69]. There are three different soluble forms of Aβ in the brain: haplotype Aβ, Aβ aggregate, and filamentous Aβ [70]. The proportion of Aβ42 content increases and accumulates in the patient’s brain to form neurofilaments and Aβ aggregates. Increasing evidence in recent years shows that Aβ aggregates are the “culprit” of this neurodegenerative disease. The deposited Aβ aggregates cause neurofibrillary tangles (NFTs) and senile plaques to form in the brain. In contrast, NFTs and senile plaques can reduce the activity of neurons and cause neurotoxic damage, leading to inflammation of the nervous system. Studies have shown that deposited Aβ aggregates can activate the expression of RAGE. Compared with non-AD patients, the expression of RAGE in hippocampal neurons, astrocytes, microglia, and endothelial cells was found to be significantly increased in AD patients [42,71]. In addition, a series of experiments have shown that Aβ can cause neurotoxicity in mice, inducing nerve apoptosis, which is enhanced by the coaction of Aβ and RAGE [43].

In experiments, Cuevas and colleagues found that after injecting Aβ into the hippocampus of rats, the expression of RAGE increased [44], and the downstream pathway protein NF-κB was also activated. Guglielmotto et al. found that two different AGEs can upregulate the expression of Aβ-secretase BACE1 by binding to RAGE through inhibitor treatment of neuroblastoma cells differentiated by SK-N-BE [72]. In Neuro-2a cells transfected with RAGE-EGFP, RAGE-Aβ can activate signal transduction pathways, including p38MAPK, and activate NF-κB and AP-1 proteins and promote the production of inflammatory cytokines IL-6, TNF-α, and macrophage colony-stimulating factor (M-CSF), eventually leading to an inflammatory response. In summary, the combination of RAGE on the nerve cell membrane and Aβ can activate NF-κB to mediate the inflammatory response and play an essential role in developing Alzheimer’s disease (Figure 4).

### 4.2. RAGE Regulates the Mechanisms Associated with Endoplasmic Reticulum Stress and Its Role in Alzheimer’s Disease

Under normal circumstances, the blood–brain barrier (BBB) can play a protective role and prevent Aβ in the blood from entering the central nervous system [45,48]. When AD occurs, the expression of RAGE on cerebral microvessels increases significantly. After injecting Aβ into the blood of AD model mice, it was found that Aβ was transported to brain tissue in a process mediated by RAGE. Following anti-RAGE action, the transport process was blocked. Through an LG-RAGE particular antibody blocking test and ap radiotracer assay, it was ultimately confirmed that Aβ penetrated the BBB in a process mediated by RAGE, a specific receptor on the vascular wall [73]. It was found that the potential mechanism of AD blood–brain barrier destruction is the activation of endoplasmic reticulum stress (ERS) in a dose-dependent manner [46].

The endoplasmic reticulum is an important location of protein processing in cells and plays a vital role in the synthesis, folding, assembly, and modification of soluble proteins and membrane proteins. When exogenous harmful substances disrupt the normal folding reaction of proteins, many abnormal proteins accumulate in the endoplasmic reticulum, triggering ERS.

Some studies have revealed that the RAGE-related signaling pathway can activate the protein kinase R-like endoplasmic reticulum kinase (PERK) pathway in ERS to control the apoptosis of human endothelial progenitor cells and affect the pathogenesis of cardiovascular disease [74]. PERK is a type I transmembrane protein that exists in the endoplasmic reticulum. It can phosphorylate the alpha subunit of eukaryotic translation initiation factor 2α (eIF2α) protein kinase. When unfolded or misfolded, proteins accumulate in the cell, and the immunoglobulin heavy-chain binding protein (BiP) is released from PERK and phosphorylates eIF2α. After phosphorylation, eIF2α can induce the expression of activating transcription factor 4 (ATF4), which activates the expression of unfolded protein response (UPR) target genes and promotes apoptosis [47]. Chen also studied the potential mechanism of the destruction of the blood–brain barrier in Alzheimer’s disease and found that Aβ42-induced RAGE activates ERS in a dose-dependent manner [45]. When cells were transfected with RAGE-siRNA, the endoplasmic reticulum stress markers decreased under the induction of Aβ42. In summary, Aβ can activate endothelial cell ERS through RAGE and increase the permeability of the blood–brain barrier. Therefore, the concentration of Aβ in the central nervous system increases, aggravating the AD condition (Figure 4).

### 4.3. Regulation of RAGE-Related Signaling Pathway by S100B Protein and Its Role in Alzheimer’s Disease

S100B is an essential member of the S100 protein family and is highly expressed in astrocytes. Under the induction of some stimuli, astrocytes can release a certain amount of S100B [49]. Through immunoblotting, EMSA ultrasonography, and reverse transcriptase-polymerase chain reaction techniques, experiments have proven that micromolar S100B can stimulate c-Jun N-terminal kinase (cJNK) phosphorylation by binding to RAGE and activating nuclear AP-1/cJun in cultured human neural stem cell transcription. In addition, Western blot, siRNA, and immunofluorescence analyses have shown that phosphorylated cJNK induced by S100B can promote the formation of NFTs in AD.

Recent studies have revealed that S100B can protect LAN-5 nerve cells at the nanomolecular level by upregulating antiapoptotic factor Bcl-2 and inhibiting the neurotoxic Aβ25–35 peptide, which can reduce the level of Bcl-2 [75]. At this concentration, S100B binds to RAGE and activates the downstream MEK-ERK1/2 pathway to stimulate the expression of survival-related genes and produce an antiapoptosis effect. After adding the MEK inhibitor, the ability of S100B to upregulate the expression of Bcl-2 was inhibited, which further confirmed the above pathway. At high concentrations, for example, due to astrocyte death, damaged astrocyte leakage, and protein clearance defects, when the intercellular concentration of S100B is expected to be higher than that of 500 nM, S100B not only does not protect LAN-5 cells from neurotoxicity from Aβ 25–35 peptide but also exerts neurotoxicity itself, which could further increase the neurotoxicity of Aβ 25–35 peptide. Furthermore, S100B was found to overactivate RAGE and its downstream ERK1/2 pathway, causing oxidative stress and inducing neuronal death. In summary, varying concentrations of S100B combined with RAGE play an opposite role in AD by activating different signal pathways. A low concentration of S100B can protect nerve cells, whereas low and high concentrations of S100B result in neurotoxicity.

### 4.4. The Mechanism of the RAGE-Related Pathway in AD

In addition to its neurotoxicity, which can activate inflammatory factors in nerve cells through RAGE and induce nerve cell apoptosis, Aβ can also combine with RAGE on the surface of endothelial cells to cause endoplasmic reticulum stress, which pathologically increases the permeability of the blood–brain barrier, increases the concentration of Aβ in the central nervous system, and further aggravates the injury of nerve cells. In addition, a high concentration of S100B was found to produce certain neurotoxicity in combination with RAGE. The combination of Aβ and S100B with RAGE shows that RAGE plays an important role in the occurrence and development of AD.

## 5. Regulatory Mechanisms of Other RAGE-Related Signaling Pathways

As a multi-ligand receptor, RAGE plays a role in immune inflammatory diseases, such as systemic lupus erythematosus and rheumatoid arthritis, by regulating the downstream NF-κB pathway and promoting the occurrence and development of diabetes, cancer, and aging. The roles of RAGE in nuclear factor E2-related factor 2 (Nrf2) pathway and autophagy are summarized in Table 2 to clarify the mechanism of RAGE in diabetes, cancer, and aging.

### 5.1. The Related Mechanism of Nrf2 Regulating the Expression of RAGE

Nrf2 is a redox regulator that plays a vital role in the physiological process of antioxidant reactions [82]. Nrf2 binds to its inhibitory cytoplasmic protein Kelch-like epichlorohydrin-related protein-1 (Keap1). When external stimuli stimulate cells, Nrf2 is dissociated from Keap1 and transported to the nucleus to bind to ARE, activating the expression of antioxidant enzyme genes to induce antioxidant stress and reduce inflammatory damage [83].

Nrf2 can regulate the activities of antioxidant enzymes and detoxifying enzymes, such as glyoxylate 1 (Glo1) [84]. Methyl glyoxal (MGO) is the primary precursor for the formation of AGEs, and the content of AGEs largely depends on the degree of transformation of MGO, whereas highly expressed and activated Glo1 can directly inhibit the formation of AGEs by catalyzing the conversion of MGO to lactic acid [76]. Numerous studies have shown that Glo1 overexpression can reduce AGE levels in diabetic model animals, inhibit the AGE–RAGE signaling pathway, and antagonize oxidative stress [85]. Some studies have shown that after intervention with the traditional Chinese medicine Eucommia ulmoides, the free Nrf2 content in the kidneys of diabetic mice increased, and the expression and activity of Glo1 protein increased significantly. In contrast, the expression of RAGE decreased [86]. AGEs and MGO are reduced considerably, proving that Eucommia ulmoides can improve the presentation of AGEs by activating the Nrf2-Glo pathway, thereby reversing kidney damage induced by the AGE–RAGE signaling pathway. In addition, experiments have proven that malondialdehyde can increase the levels of Nrf2/HO-1 and Glo and downregulate and reduce the levels of MGO to inhibit the formation of AGEs, effectively preventing inflammation and diabetes. The role of the Nrf2/Glo1-RAGE signaling pathway in inflammation, organ damage, and diabetes has attracted considerable attention [87].

Nrf2/HO-1 can also affect the interaction between HMGB1 and RAGE by inhibiting the oxidation of HMGB1, thereby regulating RAGE-related pathways. Heme oxygenase 1 (HO-1), a rate-limiting enzyme, plays an essential role in heme metabolism. HO-1 can protect mammals from inflammation and oxidative damage by inducing the production of carbon monoxide and biliverdin and their metabolite, bilirubin. Experimental studies have revealed that HO-1 can regulate the expression of RAGE by inhibiting the secretion of HMGB1. Extracellular HMGB1 is easily oxidized to form disulfide bonds between adjacent cysteine molecules 23 and 45 on A-box. Disulfide HMGB1 stimulates the production of proinflammatory cytokines by binding to RAGE receptors [77]. To antagonize the stimulation of reactive oxygen species (ROS) and other harmful factors, Nrf2 and Keap1 dissociate and activate downstream HO-1 to inhibit the oxidation of HMGB1 and indirectly regulate RAGE-related pathways. Kawahara et al. found that Ume extract, a natural triterpenoid extract from plum fruit, can inhibit the secretion of inflammatory HMGB1 by inducing the activation of the Nrf2/HO-1 pathway in LPS-stimulated mouse macrophages [88]. Western blotting analysis further confirmed that plum fruit extract can induce Nrf2/HO-1 expression and inhibit macrophage HMGB1 secretion in a concentration-dependent manner. Therefore, triterpenes can be used as a new drug for the targeted intervention in RAGE and provide a new “first aid” therapy for septicemia and other systemic inflammatory diseases caused by its related mechanisms (Figure 5).

### 5.2. Related Mechanisms of Autophagy Regulated by RAGE

Autophagy is an essential physiological process that removes damaged proteins and organelles to achieve cell metabolism and renewal of some organelles. Autophagy requires a large number of signaling pathways; the signal pathways formed between phosphatidylinositol-3-hydroxylase (PI3K)/protein kinase B (AKT), AMPK, and mammalian target of rapamycin (mTOR) play a crucial role [89,90,91]. The activated PI3K/AKT signaling pathway can activate the downstream mTOR pathway to reduce autophagy. Furthermore, AMP-dependent protein kinase (AMPK) increases the level of autophagy by inhibiting the expression of the mTOR pathway. In normal cells, RAGE binding to its ligands can reduce the expression level of the PI3K/AKT signaling pathway and activate the AMPK pathway to promote autophagy. Hou et al. found that AGEs can enhance the expression of Beclin 1 and LC3 in cardiomyocytes and increase the number of autophagy vacuoles. However, after pretreatment with RAGE-siRNA, the expression of Beclin 1 and LC3II in cardiomyocytes exposed to AGEs decreased, and the cell survival rate increased [78]. AGEs can inhibit PI3K/Akt/mTOR pathway expression through RAGE. When RAGE was specifically inhibited, the expression of PI3K/Akt/mTOR increased, whereas the level of autophagy decreased after the addition of PI3K inhibitor. This indicates that AGE–RAGE can regulate autophagy by inhibiting the expression of the PI3K/AKT/mTOR signaling pathway. SSM et al. found that after specific antibodies blocked RAGE, Aβ induced a decrease in intracellular calcium, AMPK phosphorylation, and autophagosomes, whereas overexpression of RAGE prolonged Aβ-induced AMPK phosphorylation and enhanced the transformation rate of LC3 II, indicating that Aβ-RAGE interaction can amplify the signal of autophagosome formation [79]. However, in tumor cells, RAGE is overexpressed. RAGE can activate the PI3K/AKT pathway and reduce the expression of AMPK to inhibit autophagy. Li et al. found that RAGE in hepatocellular carcinoma cells promotes the growth of tumor cells by negatively regulating the AMPK/mTOR signaling pathway, thereby inhibiting autophagy [80]. A study by Kang et al. also revealed that RAGE on the surface of pancreatic cancer cells can activate the PI3K/AKT pathway to inhibit autophagy and promote tumor growth [92]. In recent years, targeted intervention in signal pathways, such as RAGE-PI3K/AKT/mTOR and RAGE-AMPK/mTOR has provided new ideas for treating tumor diseases.

RAGE can also regulate mitochondrial autophagy by regulating the PINK1/Parkin pathway. It has been found that the binding of RAGE to AGEs on the surface of neonatal rat cardiomyocytes can activate the PINK1/Parkin pathway, and the expression of PINK1 and Parkin is significantly increased in a concentration-dependent manner. Still, the level of mitochondrial membrane potential is decreased dramatically, indicating that mitochondria are damaged [81]. In an attempt to further verify the process of mitochondrial autophagy induced by aging through RAGE, researchers found that the expression of down-regulated PINK1 and Parkin decreased in cardiomyocytes treated with mitochondrial autophagy inhibitor rings, which was consistent with the results of adding RAGE-specific antibodies. The results show that AGEs interact with RAGE to induce mitochondrial autophagy by regulating the PINK1/Parkin pathway.

Under normal conditions, RAGE inhibits PI3K/AKT and activates the AMPK/mTOR pathway to inhibit the regulatory molecule mTOR, promote autophagy, and complete the important turnover of intracellular substances, whereas overexpressed RAGE activates PI3K/AKT and inhibits the AMPK/mTOR pathway, inhibiting autophagy and promoting the growth and reproduction of cancer cells. In summary, RAGE can affect autophagy in many ways, but its mechanism is complex, so further research is needed to explore the role of RAGE in the process of autophagy.

## 6. Targeted Therapies against RAGE for the Management of Immune Diseases

RAGE knockout mice appear to be healthy and develop normally, suggesting that inhibition of RAGE is a safe therapeutic approach. Extensive studies in mice using various RAGE inhibitors, mainly excess sRAGE decoys, showed no deleterious effects of RAGE inhibition. However, in vivo administration in humans may not be the ideal approach for treating targeted RAGE. Because sRAGE is a large recombinant protein, producing therapeutically available levels would be difficult and expensive. In addition, because sRAGE is a ligand decoy, sRAGE administration prevents ligands from binding to receptors that share ligands with RAGE. Due to the deficiency of sRAGE, small-molecule inhibitors targeting the extracellular ligand binding site or, more recently, the intracellular signaling domain of RAGE have been developed.

### 6.1. TTP488 and Derivatives

TTP488 is an orally available small-molecule inhibitor of RAGE. It is also known as PF-04494700 or azeliragon, and its chemical name is 3-[4-[2-butyl-1-[4-(4-chlorophenoxy)phenyl]imidazol-4-yl]phenoxy]-*N*,*N*-diethylpropan-1-amine [93]. Although most preclinical studies on RAGE have focused on diabetic complications, cardiovascular disease, and cancer, most clinical trial work on TTP488 has been conducted in AD cohorts [93,94]. TTP488 inhibits the binding of multiple RAGE ligands, including AGEs, HMGB1, S100B, and Aβ. In a mouse AD model, TTP488 administration was found to inhibit inflammatory signaling, neuronal Aβ accumulation, and neurocognitive function [93,94]. In 2007, a large 18-month phase II trial assigned 399 subjects with mild-to-moderate AD (NCT00566397) to low-dose, high-dose, or placebo groups. The study was terminated at 6 months because the high-dose group showed worsening cognitive measures. However, a follow-up analysis of the low-dose group showed clinical benefit in slowing cognitive decline [95].

Various groups have developed new TTP488 derivatives by modifying the imidazole ring, the hydrophobic side groups, and the aromatic core. Most studies testing derivatives of TTP488 have been performed in preclinical models of AD, and inhibition of disease was observed [96,97]. None of these derivatives have progressed to human clinical studies to date.

### 6.2. FPS-ZM1

Deane et al. screened a library of 5000 small molecules to generate a new class of RAGE inhibitors for their ability to inhibit RAGE-Aβ interactions. A compound called FPS-ZM1 blocked inflammatory signaling in the mouse brain, reduced Aβ accumulation, and improved cognitive performance. Importantly, FPS-ZM1 caused no toxic side effects in mice, even at doses of up to 500 mg/kg [98].

FPS-ZM1 has been explored in other mouse models of neuropathology. FPS-ZM1 was found to reduce brain inflammation and Aβ production and improved cognitive function in a rat model of neuroinflammation with AGEs injected into the hippocampus [99]. In primary cultured rat microglia, FPS-ZM1 was found to inhibit AGE-induced inflammation and reactive oxygen species [100,101]. Recently, studies have demonstrated that small-molecule inhibitors of RAGE, including FPS-ZM1, can inhibit cancer progression and metastasis. In vitro studies with highly metastatic breast cancer cells revealed that FPS-ZM1 abrogated the excess invasion caused by RAGE. No effect was observed in terms of cell viability with FPS-ZM1, suggesting that inhibiting RAGE with FPS-ZM1 affects its migratory and invasive properties [102].

### 6.3. Other Inhibitors

Chondroitin sulfate and heparan sulfate strongly bind to RAGE and suppress lung colonization by tumor cells. Polysulfide hyaluronan GlycoMira-1111 (GM-1111) was found to inhibit interactions between RAGE and CML, HMGB1, and S100B and exhibited anti-inflammatory activity [103,104]. S100-derived peptide (ELKVLMEKEL) was found to compete for the RAGE site required to bind ligands, such as S100P, S100A4, and HMGB1, and reduced RAGE-mediated NF-κB activation, inflammation, tumor growth, and metastasis in different cancer cells [105]. In addition, peptides derived from the COOH-terminal motif of HMGB1 were also found to bind RAGE, inhibit the interaction between RAGE and HMGB1, and effectively suppress the pulmonary metastasis and invasion of tumor cells [106].

Critical issues remain to be addressed with respect to understanding RAGE-targeting therapy and the long-term impact of RAGE blockade in humans. Future investigations are required to improve understanding of the characteristics of efficient RAGE inhibitors to develop a significant understanding of the impact of RAGE blockage.

## 7. Conclusions

According to current research, the role of RAGE in immune diseases is complex, and it can bind to various ligands to play a role in many diseases. As a multimatching body on the cell surface, RAGE can regulate multiple signaling pathways and participate in various physiological and biochemical reactions when combined with the mating body, thus playing an essential role the development various immune-related diseases. In addition, RAGE is involved in a variety of signal pathways. For example, the Nrf2/RAGE signaling pathway is critical in diabetes, septicemia, and other diseases. Autophagy regulated by RAGE plays a vital role in cancer and postinjury repair and can be used as a target to monitor the occurrence and development of cancer. There are many types of RAGE-related signaling pathways, and their regulatory mechanisms are relatively complex, playing a vital role in many diseases. Probing of RAGE-related signal pathways can provide new ideas to clarify disease occurrence and development processes and provide new targets for diagnosis and treatment of diseases. Therefore, there is much to be explored and clarified with respect to the RAGE-related signaling pathway and its mechanism in various conditions.

## Figures and Tables

**Figure 1 molecules-27-04922-f001:**
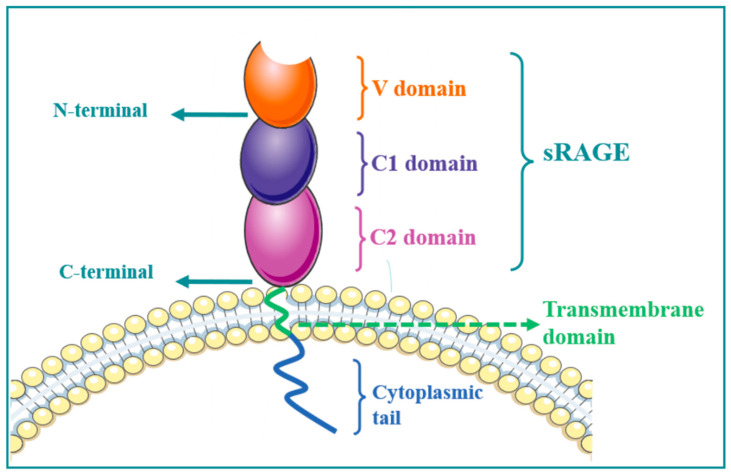
RAGE structural organization. The extracellular domain comprises three domains: V, C1, and C2. One transmembrane receptor passes through the plasma membrane bilayer, followed by an intracellular cytoplasmic tail. The extracellular region without a transmembrane receptor and a cytoplasmic tail is called soluble RAGE (sRAGE).

**Figure 2 molecules-27-04922-f002:**
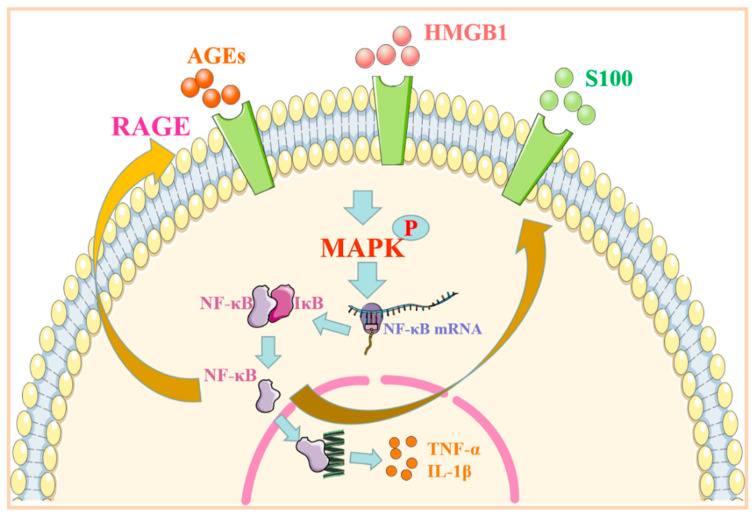
RAGE regulates the MAPK/NF-κB signaling pathway. After binding to the ligand, RAGE phosphorylates its downstream MAPK and activates NF-κB protein. NF-κB enters the nucleus to promote the transcriptional expression of inflammatory factors. Abbreviations: AGEs: advanced glycation end products; HMGB1: high-mobility group protein 1; RAGE: receptor for advanced glycation end products; MAPK: mitogen-activated protein kinase; NF-κB: nuclear factor kappa-B; TNF-α: tumor necrosis factor α; IL-1β: interleukin-1β.

**Figure 3 molecules-27-04922-f003:**
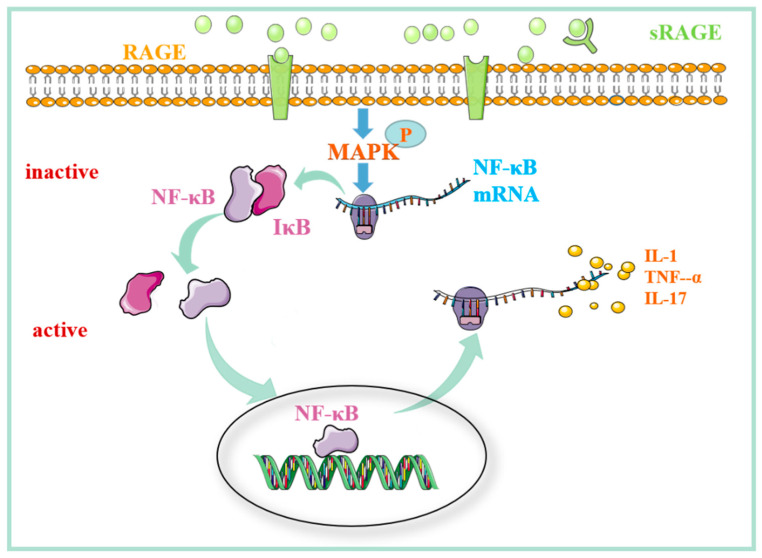
The mechanism of action of RAGE–ligand binding in RA. The binding of RAGE with the ligand activates the MAPK/NF-κB signaling pathway and induces the production of inflammatory factors. sRAGE can block the ligand–RAGE interaction on the cell surface, reducing the entry of inflammatory cells into the joint cavity. Abbreviations: RAGE: receptor for advanced glycation end products; sRAGE: soluble receptor for advanced glycation end products; MAPK: mitogen-activated protein kinase; NF-κB: nuclear factor kappa-B; IκB: inhibitor of NF-κB; TNF-α: tumor necrosis factor α; IL-1: interleukin-1; IL-17: interleukin-17.

**Figure 4 molecules-27-04922-f004:**
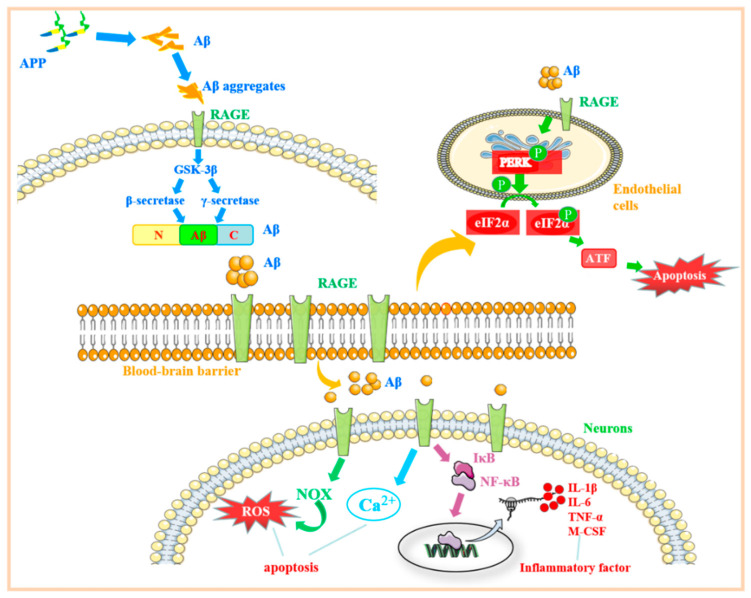
RAGE-related signaling pathways in Alzheimer’s disease. After APP is converted to Aβ, on the one hand, it combines with RAGE to activate the downstream MAPK/NF-κB pathway to release inflammatory factors and damages nerve cells. On the other hand, Aβ binds to RAGE on the surface of BBB cells, causing ERS, altering the permeability of the BBB, increasing the concentration of Aβ in the central nervous system, and aggravating AD-related symptoms. Abbreviations: APP: amyloid prebiotic protein; Aβ: amyloid β-protein; RAGE: receptor for advanced glycation end products; GSK-3β: glycogen synthase kinase 3β; PERK: protein kinase R-like endoplasmic reticulum kinase; eIF2α: eukaryotic translation initiation factor 2α; NOX: nitrogen oxide; NF-κB: nuclear factor kappa-B; IκB: inhibitor of NF-κB; TNF-α: tumor necrosis factor α; IL-1β: interleukin-1β; IL-6: interleukin-6; M-CSF: macrophage colony-stimulating factor.

**Figure 5 molecules-27-04922-f005:**
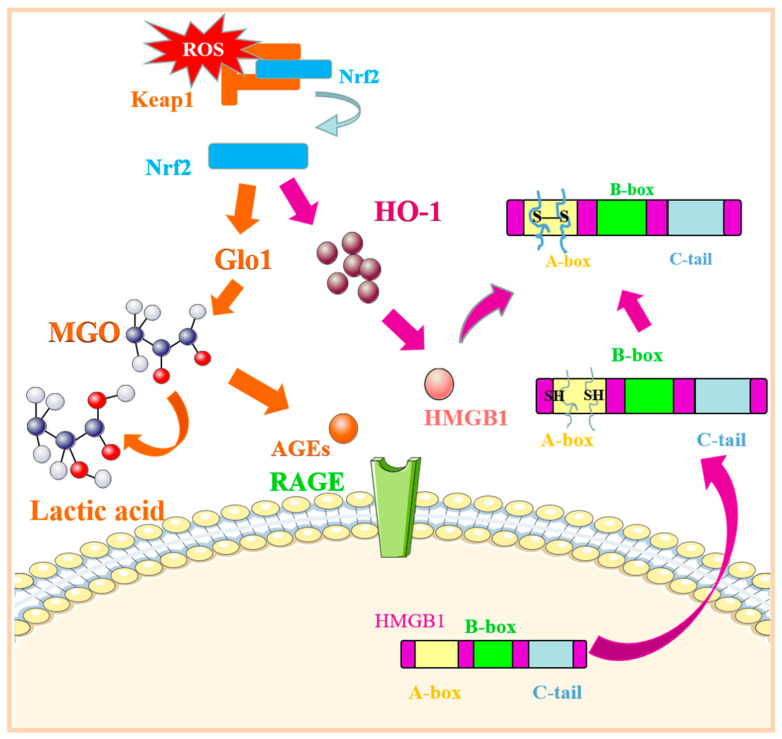
Nrf2 regulates the related pathways of RAGE expression. After being stimulated by oxidative stress and other stimuli, Nrf2 breaks away from keap1 and deactivates Glo1, which can reduce the production of AGEs, thereby regulating the level of RAGE expression. Similarly, after Nrf2 is activated, it can activate its downstream HO-1 to exert its antioxidant effect, inhibit the oxidation of HMGB1, and regulate the expression of RAGE. Abbreviations: ROS: reactive oxygen species; Nrf2: nuclear factor E2-related factor 2; Keap1: Kelch-like epichlorohydrin-related protein-1; Glo1: glyoxylate 1; HO-1: heme oxygenase 1; MGO: methyl glyoxal; AGEs: advanced glycation end products; HMGB1: high-mobility group protein 1.

**Table 1 molecules-27-04922-t001:** The role of RAGE-related signaling pathways in immune-associated diseases.

Signal Pathway	Ligand	Specificity	Reaction	Diseases	References
HMGB1/RAGE-MAPK/NF-κB	HMGB1, AGEs	Endothelial cells	Production of inflammatory and B-cell-activating factors ↑	SLE	[37]
		Immune cells	Inflammatory reaction ↑		[38]
		Synovial cells	Inflammatory factor ↑	RA	[39]
S100/RAGE-MAPK/NF-κB	S100A12	Epithelial cells	EMT ↑	Pulmonary fibrosis	[40]
	S100A9	Fibroblasts	Cell proliferation and secretion of inflammatory factors ↑		[41]
Aβ/RAGE-MAPK/NF-κB	Aβ	Neurons	NFTs ↑ Cell activity ↓ Inflammatory factor ↑	AD	[42,43,44]
Aβ/RAGE-ERS	S100B	BBB	ERS ↑ Permeability of blood–brain barrier ↑	AD	[45,46,47]
S100B/RAGE-MEK/ERK1/2		Neurons	Low concentrations of S100B protect nerve cells; very low and high concentrations of S100B produce neurotoxicity		[48,49]

**Table 2 molecules-27-04922-t002:** Regulatory mechanisms of other RAGE-related signaling pathways and typical diseases involved.

Signal Pathway	Ligand		Changes in Cell Signaling	Typical Disease	References
Nrf2/Glo1	AGEs	Upstream	Glo1 catalyzes the conversion of MGO to lactic acid, AGEs ↓	Diabetes	[76]
Nrf2/HO-1	HMGB1		HO-1 inhibits the desulfurization of HMGB1	Inflammatory diseases	[77]
PI3K/AKT AMPK/mTOR	AGEs, Aβ	Downstream	Normal cell: AMPK/mTOR ↑PI3K/AKT ↓ Autophagy ↑ Tumor cell: AMPK/mTOR ↓ PI3K/AKT ↑ Autophagy ↓	Cancer	[78,79,80]
PINK1/Parkin	AGEs		PINK1/Parkin ↑ Mitochondrial autophagy ↓	Senescence	[81]

## Data Availability

No new data were created or analyzed in this study. Data sharing is not applicable to this article.

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
