# Peer review of "Receptor for Advanced Glycation End Products (RAGE): A Pivotal Hub in Immune Diseases"

_molecules, 2022, doi:10.3390/molecules27154922_

Round 1

Reviewer 1 Report

The authors summarize evidence for the involvement of the receptor for advanced glycation end products (RAGE) in triggering inflammation with special emphasis on the pathogenesis of several immune disorders. The authors provide a detailed and organized review where they list prior studies on RAGE structure, its link to the MAPK/NF-κB signaling pathway, and regulation of the Nrf2/HO-1 pathway and autophagy. Then, the authors describe the role of RAGE in immune-associated disorders and Alzheimer’s disease. The authors provided four interesting and informative figures that summarize the regulation of the MAPK/NF-κB signaling pathway (Figure 1), mechanism of action of RAGE-ligand in RA (Figure 2), RAGE signaling in Alzheimer’s disease (Figure 3), and RAGE expression regulation by Nrf2 signal (Figure 4).  

Comments:    

1) In order to attract the interest of more readers regarding the clinical relevance of the work, an additional section is suggested to be added to address the targeted therapies against RAGE for the management of immune diseases. This may include small molecule inhibitors.

2) The provided sections read like narration for the evidence of discussed points without critical aspects/reflection points. At the end of each section, a take-home message is advised to be provided.

3) The authors are advised to make the figure caption stand-alone. To this end, authors are advised to provide the full names of all the listed abbreviations in the figures. Please, address this issue in all listed figures.

4) The authors are advised to avoid “future tense; will” in figure legends in the current review and replace them with the “present tense”.

For example, in the legend of figure 4 (lines 405-407), the authors state that “After being stimulated by oxidative stress and other stimuli, Nrf2 will break away from keap1 and deactivate Glo1, which can reduce the production of AGEs, thereby regulating the level of RAGE expression”. Herein, please consider replacing “will break” with “breaks”. Please, address this issue in the entire manuscript.

5) Some typos/syntax errors are present in the manuscript which need to be addressed, for example:

- In lines 13-14, the authors state that “Recently, increasingly studies have shown that RAGE ligand binding …….”.

Please, consider correcting the above statement to “Recently, increasing studies have shown that RAGE ligand binding …….”.

- In line 217: the authors state that: “The binding of RAGE and ligand activates the MAPK/NF-κB signaling pathway ….”

Please, consider correcting the above statement to “The binding of RAGE with the ligand ……”.

- In line 219: the authors state that: “SRAGE can block. …”.

Please, consider correcting “SRAGE” to “sRAGE”.

6) To avoid readers’ confusion, some statements need to be elaborated/carefully revised. For example, in lines 219-220, the authors state that “SRAGE can block the ligand-RAGE interaction on the cell surface, reducing the entry of inflammatory cells into the joint cavity”.

Please, consider clarifying the sentence for example by “By competitively binding RAGE ligands, sRAGE lowers the entry of inflammatory cells into the joint cavity”.

7) To make Table 1 clearer and more informative to readers, authors are advised to add a visual distinction that separates each signaling pathway and its ligands from the next one.

Please, address the same issue in table 2.

8) More recent 2011-2022 references are advised to be added to the review.  

Reviewer 2 Report

Here are the comments that in my opinion may help to improve the current manuscript:

In lines 38-41, please include references for the statements of receptor expression and putative functions of the receptors, if possible including some brief examples. It would also be nice to differentiate the role of RAGE in physiological and pathological conditions.

Lines 43-45: the authors state that RAGE signaling is indispensable for the occurrence and development of some immune diseases. It would be nice to include a brief explanation of why RAGE signaling is indispensable of the development of psoriasis or systemic lupus erythematosus.

Lines 54-65: On RAGE structure and receptor types (transmembrane type and soluble-type with antagonist functions) would be nice to include an illustrative figure.

Line 67: For RAGE polymorphisms, it would be nice to include some illustrative examples in the text.

Lines 68-69: On the ligands, no AGEs are included and some new others are mentioned. It would be nice to homogenize across the manuscript.

Line 75: the authors state “AGEs are brown”; it is not clear to me what this means.

Lines 76-78: please rephrase this sentence, I cannot understand what the author’s point is.

Line 92: the authors state “when stimulated by the outside world”; please, rephrase the sentence; the scientific english language style is not correct. With a “when stimulated” or “when stimulated by extracellular ligands” would be enough.

Lines 99-100: please indicate what an EF hand like structure consists of.

Legend figure 1: it would be nice to add the effect of the positive feedback loop between intracellular RAGE signaling and subsequent ligand binding (which is, illustrated in the figure).

Line 198: please indicate what seronegative/seropositive RA mean.

Lines 199-201: I suggest moving the sentence of the role of sRAGE in SLE to the SLE section. It is a little weird to change to SLE in the RA section and then go back to RA.

Line 224: the authors state “epitelial cells, lymphocytes etc”, in my opinion the “etc” is not scientifically very accurate. Please include all information or the most important information and indicate that there are others of which you are not going to talk about.

Line 314: the authors state that “a-beta activates endoplasmic reticulum stress in nerve cells and disrupt the blood brain barrier”. However, I am not sure whether what they explain above relates to neurons, endothelial cells, or both.

The legend of figure 3, in my opinion, is not complete. I suggest the authors explaining briefly in the legend, all what the figure illustrates.

Lines 352-358: please rephrase the paragraph, it is very difficult to understand what the authors want to explain.
